# Design and Thermal Analysis of Flexible Microheaters

**DOI:** 10.3390/mi13071037

**Published:** 2022-06-29

**Authors:** Dezhao Li, Yangtao Ruan, Chuangang Chen, Wenfeng He, Cheng Chi, Qiang Lin

**Affiliations:** 1Zhejiang Provincial Key Laboratory of Quantum Precision Measurement, Collaborative Innovation Center for Information Technology in Biological and Medical Physics, College of Science, Zhejiang University of Technology, Hangzhou 310023, China; yangtaoruan@gmail.com (Y.R.); chuangang_chen@yeah.net (C.C.); wenfeng_he1518@yeah.net (W.H.); 2Key Laboratory for Thermal Science and Power Engineering of Ministry of Education, Department of Engineering Mechanics, Tsinghua University, Beijing 100084, China; cchi@connect.ust.hk

**Keywords:** microheater, heating wire structure, PID control, graphene thin film, fast response

## Abstract

With the development of flexible electronics, flexible microheaters have been applied in many areas. Low power consumption and fast response microheaters have attracted much attention. In this work, systematic thermal and mechanical analyses were conducted for a kind of flexible microheater with two different wire structures. The microheater consisted of polyethylene terephthalate (PET) substrate and copper electric wire with graphene thin film as the middle layer. The steady-state average temperature and heating efficiency for the two structures were compared and it was shown that the S-shaped wire structure was better for voltage-controlled microheater other than circular-shaped structure. In addition, the maximum thermal stress for both structures was from the boundary of microheaters, which indicated that not only the wire structure but also the shape of micro heaters should be considered to reduce the damage caused by thermal stress. The influence resulting from the thickness of graphene thin film also has been discussed. In all, the heating efficiency for flexible microheaters can be up to 135 °C/W. With the proposed PID voltage control system, the response time for the designed microheater was less than 10 s. Moreover, a feasible fabrication process flow for these proposed structures combing thermal analysis results in this work can provide some clues for flexible microheaters design and fabrication in other application areas.

## 1. Introduction

Compared with a hotplate or Peltier elements, microheaters with thermal mass and power consumption reduced, are more effective for portable applications [1]. Microheaters have been widely applied in various applications including gas sensors [2,3], microcalorimeters [4], gas flow meters [5], infrared sources [6,7], and thermal management [8]. Various structure designs of microheaters have been proposed for different applications. Zhou, Q. et al. designed a floating microheater composed of 72 heating units based on semiconductor oxide (SMO), and the total size was 3.2 mm × 3.0 mm [9]. Solzbacher, F. et al. designed a suspended gas chamber micro heating plate using metal oxide for the gas sensor [10]. The size of the heating film was 100 μm × 100 μm with the heating wires’ shape and width optimized. The reported maximum heating temperature of the heating film was 700 °C. Krishna, R.M. et al. designed an integrated photonic resonant wavelength tuning device based on a polysilicon microheater, which has a wide tunable range and can be applied to provide efficient and high-speed tuning wavelength for resonant devices [11]. Cho, J. et al. proposed a microheater for carbon monoxide (CO) detectors to enhance the sensitivity by more than 20%, and reduce response time to less than half [12].

Moreover, as the essential component in gas sensors [13] and biological sectors [14], different kinds of research about the structures and materials of microheaters have been done. Waghmare, S. et al. optimized the temperature uniformity and stability of the microheater considering the influences of the wire structure and applied voltage with COMSOL Multiphysical software [15]. Hasan, M.H. et al. discussed a microheater with platinum (Pt) for gas sensing applications using COMSOL Multiphysics [16]. Prajesh, R. et al. depicted the benefit of low thermal conductivity of substrate by comparing three different rigid materials (silicon, alumina, and glass) [17]. Tiwari, S.K. et al. demonstrated a flexible microheater with Polyethylene terephthalate (PET) substrate and the influence of supply voltage and time on heater temperature profile was analyzed using COMSOL Multiphysics software [18].

In this paper, we proposed a type of flexible microheaters combining a classical wire design and graphene sheet for gas sensing applications in wearable devices. Recently, with the development of flexible electronics, low power consumption and fast response microheater are desired in many areas [6,19]. Because of the unique electrical and thermal conductivity, the graphene-based electrothermal heater has shown the properties of fast response, flexibility, and high-efficiency energy conversion [20,21,22,23]. Graphene-based typical structures fabricated with different methods are shown in Figure 1. However, limited by fabrication techniques of graphene, it is not convenient to get the required wire. Combined with a classical wire design with a graphene sheet, we proposed a structure type for a flexible microheater. For low-power consumption and fast response flexible microheater design, different substrate material, wire structure, and electric control system influences were discussed in this paper. These results can provide guidelines to get high-efficiency flexible microheaters for different applications.

## 2. Models and Methods

Heat conduction, heat convection, and heat radiation are the three different heat transfer types between the chip and the air layer involved in the working process of a microheater. However, their effects on heater performance are varied.

### 2.1. Heat Transfer Models

To describe the electric heating and heat transfer process of the microheater, we applied the following formula for this model.
(1)$$\begin{array}{c} \rho C_{P}u\nabla T=\nabla \cdot \left(k\nabla T\right)+Q_{\varepsilon }\\ Q_{\varepsilon }=J\cdot E \end{array}$$
where ρ  is the density of the material, $C_{P}$ is the constant pressure-specific heat of the material,  k is the thermal conductivity coefficient, $Q_{\varepsilon }\;$ is the Joule heating power, J is the density surface current, E is the applied voltage.

Heat conduction has a significant influence on the performance of microheaters, Fourier’s law was applied to quantitatively describe the heat conduction process:(2)$$q_{x}^{’}=-k\frac{\partial T}{\partial x}$$
where $q_{x}^{’}\;$ is the heat flux density,  k is the thermal conductivity,  T is the absolute temperature; x is the heat conduction distance. The thickness of the substrate is the key parameter for the heat conduction process following the Fourier law. Under the same condition, the chip temperature will change significantly as long as the thickness of the substrate is adjusted. In this study, we set the material and size of the substrate to optimize the thickness.

For convection without considering external interference, Newton’s law can be applied as follow.
(3)$$q=hA\left(T_{w}-T_{f}\right)$$
where q is the convection heat flux, h is the convective exchange coefficient, A is the area of the heat exchange area, $T_{w}$ is the solid surface temperature, $T_{f}$ is the fluid temperature. In this study, convection mainly occurs between the air and the microheater, we set the thermal convection coefficient as 5 W/(m^2^⋅K).

Thermal radiation refers to the thermal energy radiated by objects. we used the following formula to describe the radiation process.
(4)$$\varphi =\varepsilon_{1}\delta A_{1}\left(T_{2}^{4}-T_{1}^{4}\right)$$
where ε  is the emissivity of the object, σ is the Stefan- Boltzmann constant which is 5.67 × 10^−8^ W/(m^2^⋅K^4^), $A_{1}$ is the radiation surface area, $T_{2}$ is the surface temperature of the object and $T_{2}$ is the ambient temperature. Through preliminary analysis, the thermal radiation power of the design target in this study is around 10^−5^ W, which is negligible compared with two the other heat transfer models having power levels of around 10^−3^ W.

### 2.2. Materials

To get a low-power consumption, high-efficiency flexible microheater, we proposed a type of structure using Polyethylene terephthalate (PET) as substrate. Due to its anisotropic thermal conductivity [24], graphene thin film was applied to enhance the temperature uniformity of the microheater. We applied copper wire for the electric heater. The material properties used in this study are summarized in Table 1.

In this study, COMSOL Multiphysics 5.5 was used to analyze the performance of the flexible micro heater, which can solve Maxwell’s equations, magnetic field equations, and boundary conditions simultaneously with the finite element method [25,26]. All the abbreviations used in this paper are summarized in Table 2.

## 3. Results and Discussion

### 3.1. Structure Design and Analysis

Different wire geometries applied for microheaters have been widely discussed [16,27,28,29]. It was reported that less inner thermal stress was generated with circular-type wire structures [1]. In this study, two typical structures with different wire geometries selected for flexible microheaters were discussed, as shown in Figure 2.

In this study, the thickness of the substrate (*t*) was 150 μm, and the sample size was 1 cm × 1 cm. To compare the heating efficiency resulting from different wire structures, circular-shape, and S-shape structures were discussed as shown in Figure 1. For these wire structures, the width of the wire (*w*) was fixed as 200 μm and the distance (*d*) was 700 μm. Additionally, we fixed the thickness of the graphene thin film for both structures as 50 nm and the input voltage 1.5 V. Furthermore, we set the default temperature for the simulation analysis as 20 °C. To set up the finite element methods (FEM) analysis mode, multi-physical field coupling modules including “structural mechanics”, “heat transfer” and “AC/DC” modules were considered simultaneously with COMSOL Multiphysics 5.5. The air environment with the heat transfer coefficient of 5 W/(m^2^⋅K) was set as the boundary condition of microheaters. The sweep mesh method was used for microheaters and free tetrahedron mesh was used for the other regions. The surface temperature distribution of the two shape structures was simulated as in Figure 3. The steady-state average temperature of the S-shape structure, shown in Figure 3a, was higher than that of the Circular-shaped structure, shown in Figure 3b, under the same input voltage condition. The higher average temperature should result from its lower electrical resistance.

With different input voltages, we evaluated the heating performance of the two structures as in Figure 4a. It was shown a non-linear quadratic relation between the steady-state average temperature and input voltage. Considering the same initial temperature *T_ini_* in this model, the average temperature *T_fin_* of microheaters results from the Joule heat $Q_{in}$ generated by electrical power and heat dissipation $Q_{out}$ to environment. Meanwhile, $Q_{in}$ can be expressed as in Equation (6) which is quadratically related to the input voltage. These results indicated that the average temperature of microheaters is mainly influenced by input electrical power other than the heat dissipation part. We found that the resistance of the Circular-shaped structure was 92.4 Ω, which was 25.0 Ω of the S-shaped structure for the same size substrate. The reason is that the total length of the Circular-shaped structure wire is longer than that of the S-shaped structure wire under the same condition.
(5)$$Q_{in}-Q_{out}=Cm\left(T_{fin}-T_{ini}\right)$$
(6)$$Q_{in}=\frac{V_{in}{}^{2}}{R}\int tdt$$
where *C* is the specific heat capacity of the microheater, and *m* is the mass of the microheater.

To compare the heating efficiency of the two structures, we proposed a factor named heating efficiency to get a quantitative analysis which was calculated as below:(7)$$\eta =\frac{\Delta T}{P}$$
where η is heating efficiency, ΔT is the temperature increased under the input electrical power of *P.* The heating efficiency of microheaters with two types of wire structures was calculated as in Figure 4b. The heating efficiency of the Circular-shaped structure was slightly lower than that of the S-shaped structure, which should be caused by higher heat dissipation with a longer heating wire. Based on the analysis results, it is better to choose an S-shaped structure for voltage-controlled heater design other than a Circular-shaped wire structure.

The main stress of this type of microheater is caused by inner thermal stress between different material layers. To further evaluate the influence of different wire structures on the heater’s internal thermal stress, we compared their internal thermal stress under the same steady-state average temperature for both structures with different input voltages. The simulated internal stress distribution was calculated and shown in Figure 5. The greatest stress for both structures was from the edges of the device which should be carefully considered to construct a robust flexible heater. These simulation results provided a clue that to reduce the damage caused by thermal stress, not only the heating wire structure but also the shape of the microheater should be considered. In addition, we also noticed the S-shaped structure showed less generated internal stress compared with the Circular-shaped and it should be better to choose this structure for square type microheater.

Usually, temperature uniformity is a key factor to evaluate the performance of a heater, we choose the temperature distribution along the center line to get a quantitative evaluation. As shown in Figure 6, the temperature uniformity of the S-shaped structure in Figure 6a is better than that of the Circular-shaped in Figure 6b. We noticed that there was temperature fluctuation of the Circular-shaped wire structure along the center line. In addition, these fluctuation-generated positions were exactly the positions of the heating wires. Considering the shape of the Circular-shaped wire structure, the fluctuation should result from the heat accumulation near the heating wire with large curvature.

In summary, we found that for voltage source powered square-type microheater design, it is more appropriate to use an S-shaped wire structure to get a better heating efficiency, less internal thermal stress, and better temperature uniformity compared with the Circular-shaped wire structure.

Other than the wire shape design, the thickness of the graphene thin film also has a certain influence on the heating efficiency of the microheater. The steady-state average temperature for the S-shaped structure corresponding to different voltage and different thickness of the graphene thin film was evaluated as in Figure 7. Graphene thin film can be beneficial to the heating efficiency of the microheater, but they are not a linear relationship.

For the real application of the proposed microheaters, we proposed a feasible preparation method for the designed microheaters. The preparation process flow was shown in Figure 8. With a suitable PET substrate as in Figure 8a, low-pressure chemical vapor deposition (LPCVD) can be applied for graphene middle layer deposition as in Figure 8b. After that, copper wire can be directly deposited with the evaporation method using a shadow mask as in Figure 8c. Finally, the proposed microheater can be fully fabrication as in Figure 8d.

### 3.2. Proportional-Integral-Derivative (PID) Control for Flexible Microheater

To reach the set temperature as soon as possible, we designed the PID control model with COMSOL software. For the PID control model, the input voltage was the controlled parameter and the average temperature of the heater was applied as a feedback factor. With an optimized PID control model for a microheater with an S-shaped structure and 50 nm thickness of graphene thin film, we can get the target temperature within 10 s which is more than 60 s without a PID control model as in Figure 9. The optimized PID values would be related to the heat dissipation environment which should be designed for real application.

## 4. Conclusions

In this paper, we proposed structure designs for flexible microheaters. Polyethylene terephthalate (PET) was chosen as the substrate material with a graphene thin film middle layer. Moreover, S-shaped and circular-shaped copper heating wire structures were proposed and systematically analyzed. The size of the heater unit was 1 cm × 1 cm, which can achieve heating efficiency up to 135 °C/W. Compared with the S-shaped wire structure, under the same wire width and space distance, the total resistance of the circular-shaped wire structure was higher. Meanwhile, the average temperature of a microheater with an S-shaped structure was higher with the same input voltage. In addition, the heating efficiency of the Circular-shaped structure was slightly lower than that of the S-shaped structure, which may be caused by higher heat dissipation with a longer heating wire. The thermal stress of two-wire structures under the same temperature was also calculated. It was shown that the maximum thermal stress from the boundary of microheaters. Moreover, we noticed there was temperature fluctuation of the Circular-shaped wire structure along the center line. Considering the shape of the Circular-shaped wire structure, the fluctuation should result from the heat accumulation near the heating wire with large curvature. Based on the internal stress and heating efficiency analysis results, it should be better to choose an S-shaped wire structure for a square-type microheater. The influence resulting from the thickness of graphene thin film was discussed and it was shown that graphene thin film can be beneficial to the heating efficiency of the microheater. In addition, a PID control system was also discussed to get a fast heating response and the analysis results indicated the response time was less than 10 s for the targeted temperature of 80 °C.

## Figures and Tables

**Figure 1 micromachines-13-01037-f001:**
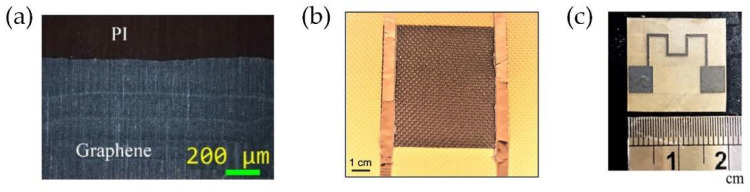
Structures of microheaters based on graphene fabricated with (**a**) inkjet-printed method [21], (**b**) laser-induced method [22], and (**c**) ultrafast laser ablation method [23]. Reprinted with permission from MDPI, ACS, and ELSEVIER.

**Figure 2 micromachines-13-01037-f002:**
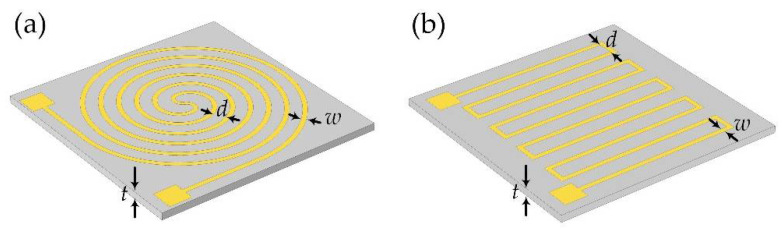
Two kinds of heating structures were studied in this study, (**a**) the Circular-shaped structure, and (**b**) the S-shaped structure.

**Figure 3 micromachines-13-01037-f003:**
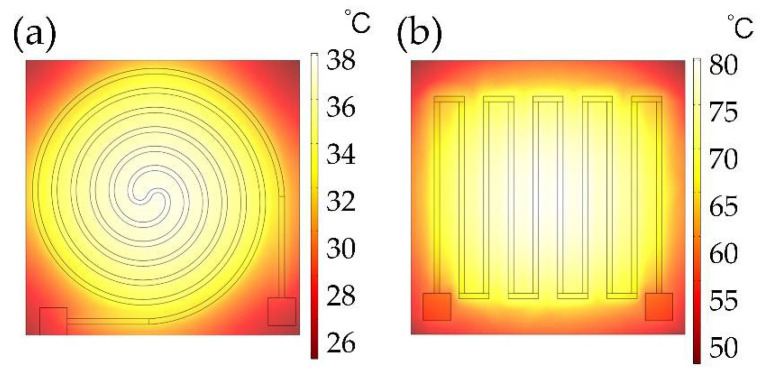
The temperature distribution of (**a**) Circular-shaped structure, and (**b**) S-shaped structure in steady-state.

**Figure 4 micromachines-13-01037-f004:**
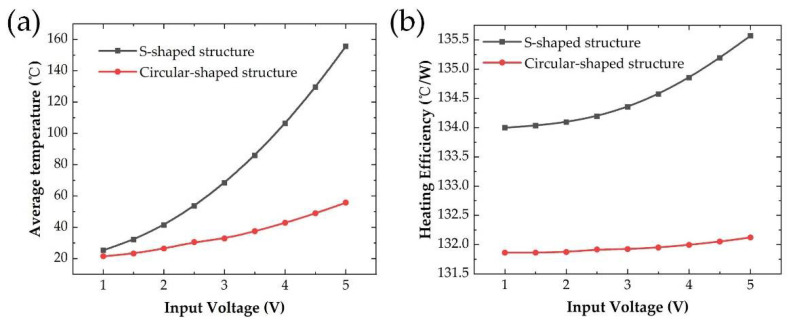
(**a**) The steady-state average temperature, (**b**) Heating efficiency for the S-shaped and the Circular-shaped structure with different input voltage.

**Figure 5 micromachines-13-01037-f005:**
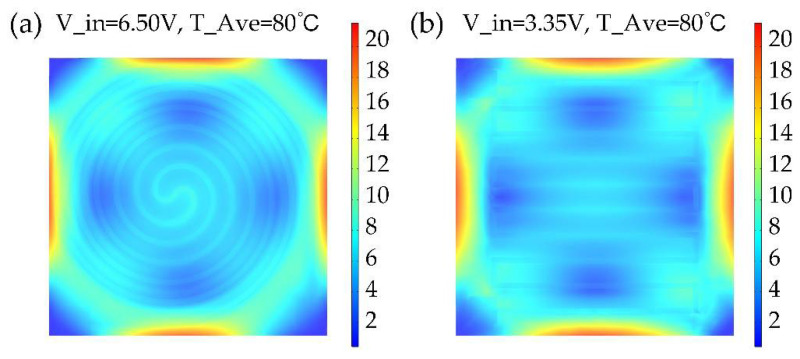
The internal thermal stress distribution of (**a**) Circular-shaped structure, and (**b**) S-shaped structure with the same steady-state average temperature of 80 °C.

**Figure 6 micromachines-13-01037-f006:**
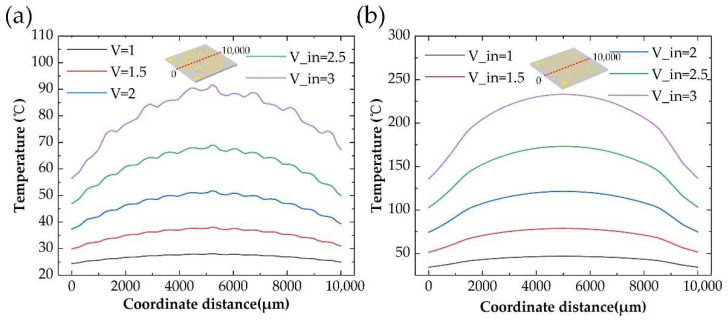
The surface temperature distribution along the center line for (**a**) the Circular-shaped structure, and (**b**) the S-shaped structure with different input voltages, with structure indicated as in the inserted picture.

**Figure 7 micromachines-13-01037-f007:**
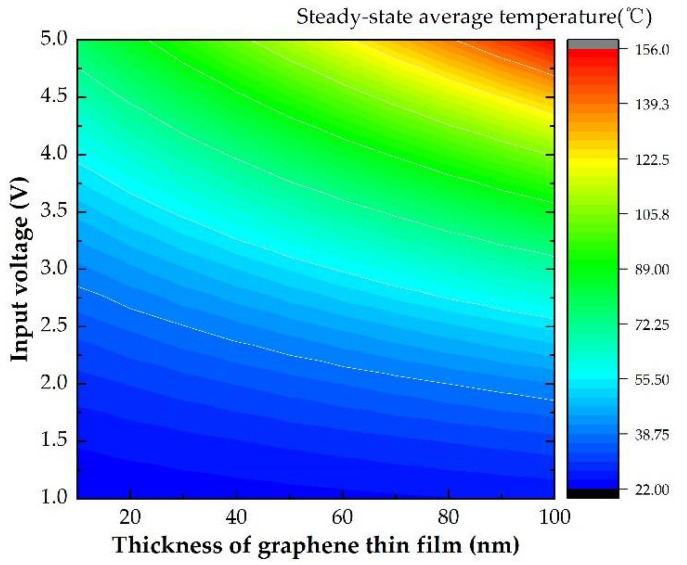
The average temperature of microheater with different input voltage for different thicknesses of graphene thin film.

**Figure 8 micromachines-13-01037-f008:**
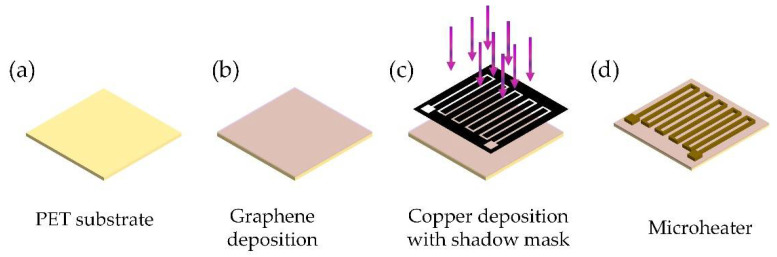
The feasible fabrication process flow for proposed microheaters, (**a**) PET substrate, (**b**) Graphene deposited with LPCVD, (**c**) Copper heating wire structure deposited with shadow mask, (**d**) The fabricated microheater.

**Figure 9 micromachines-13-01037-f009:**
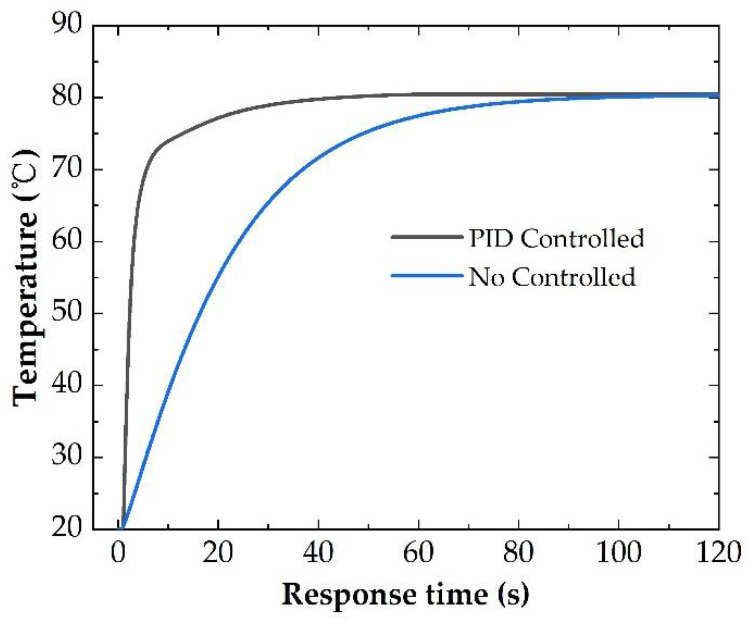
The response time of the microheater with and without PID control.

**Table 1 micromachines-13-01037-t001:** Material properties used in this study.

Material	PET	Graphene [24]	Copper
Young’s modulus (Pa)	4.00 × 10^9^	1.00 × 10^12^	1.278 × 10^10^
Poisson’s ratio	0.125	0.160	0.326
Thermal expansion coefficient (1/K)	3.30 × 10^−5^	2.90 × 10^−5^	1.890 × 10^−5^
Thermal Conductivity (W/(m⋅k))	0.14	x: 3000, y: 3000, z: 6.1	380
Constant pressure heat capacity (J/(kg⋅K))	1100	1365	390
Density (kg/m³)	1370	2330	8960

**Table 2 micromachines-13-01037-t002:** List of abbreviations used in this paper.

Abbreviations	Meaning
SMO	semiconductor oxide
CO	carbon monoxide
MEMS	micro-electromechanical Systems
LPCVD	low-pressure chemical vapor deposition
PET	polyethylene terephthalate
Pt	platinum
PID	proportional-integral-derivative

## Data Availability

The data presented in this study are available on request from the corresponding author.

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
