# Peer review of "Design and Thermal Analysis of Flexible Microheaters"

_micromachines, 2022, doi:10.3390/mi13071037_

Round 1

Reviewer 1 Report

The author mainly described the experimental process of the flexible micro-heaters with S-shaped wire structure and the Circular-shaped wire structure, and recorded the corresponding test results. But the lack of reasonable explanation for the experiment phenomenon made the manuscript lose its research significance.

For example, why is the steady-state average temperature of the S-shape structure higher than that of the Circular-shaped structure? etc.

Author Response

We thank the reviewer for the careful examination and the constructive comments. To improve our manuscript, we have revised the manuscript accordingly to address the issues raised by the reviewer. All the revised and added parts have been marked in red in the manuscript.  The point-by-point responses to the reviewer’s comments as in the attached document.

Reviewer 2 Report

This paper presented some COMSOL simulation results for two microheater designs with an embedded graphene layer. Although the results might be of interest to some researchers in the relevant field, I believe that they are not significant enough to be published as a journal paper. Therefore, I would not recommend it for publication, and suggest the authors to consider submitting it to a conference or as a letter instead.

In addition, the English language needs to be revised, as some articles (‘a’, ‘the’) are missing and some sentences need rephrasing. A more detailed background study is necessary as a supporting evidence to show the novelty and advantages of the proposed microheater designs.

To publish this paper as a journal article, I would suggest the authors to consider the fabrication process of the device. An analysis of the possible approaches is preferred, or even better, an actual prototype. Some microscopic images or experimental measurements will greatly improve the quality of the paper as a journal publication. 

Author Response

(The authors gave the same response as above.)

Reviewer 3 Report

This work investigates the performance of the flexible micro-heater with different shapes including circular and s-shape. The idea and implementation of a graphene-based heater are novel, however, this work can improve in certain directions in order to achieve this study in a practical way. The manuscript has an impact on the MEMS field however I have the following concerns and would like to receive answers from the authors.

- The introduction is not sufficient and does not cover most of the works in this field. 

- The scope of the paper is not clear.

- S-shaped micro-heater is not new. What is your novelty here? Please refer to the following paper (Simulation of low power heater for gas sensing application)

-Add a list of abbreviations. 

- Full details about the FEM model are required such as module, mesh, study ....etc.

- Fig.3 shows non-linear quadratic relation between the steady-state average temperature and input voltage.  Why? Is it something related to the stiffness?

- There is some noise or disturbance associated with surface temperature distribution along the center line of the circular shape. Any clarification? 

- Have you studied the effect of graphene layer thickness on the overall performance of the micro-heater? 

- PID controller helps reach the set temperature faster, but this will heat up the device and could lead to damage. At which thickness level was this analysis?

- Which fabrication process is going to be used to fabricate the graphene sheets as a micro-heater? This is a hot topic.

-Several typos need to be addressed.

Author Response

(The authors gave the same response as above.)

Round 2

Reviewer 1 Report

In Equation (5), the author should point out what the symbols C and m denote respectively?

Author Response

Dear Reviewer:

We thank you for the careful examination and these constructive comments! We have revised the manuscript accordingly to address these raised issues.  All the modifications have been done in “Track changes mode” following the instruction by Editor. 

Reviewer 2 Report

Dear Authors,

Thank you for revising the manuscript and providing a cover letter. I agree that theoretical simulations are of interest to many researchers, me included. Because of that, I would like to see either a more comprehensive simulations with multiple variables, like the papers referred in the cover letter, or a link between simulations and real-world devices. Therefore, I am glad to see the authors have considered my advice and add some fabrication processes to the manuscript.

Before submitting the final version, I would advice the authors to revise the English again, and reorganise the introduction. At the moment, the examples listed in the introduction are a bit random. They can be categorised in a more logic way, such as by materials, by designs, by applications, etc. I personally would expand the introduction a bit further, perhaps even adding a paragraph dedicated to graphene on microheaters.

Overall, I would recommend this paper for publication.

All the best!

Author Response

(The authors gave the same response as above.)

Reviewer 3 Report

Dear Authors, 

Many thanks for the revised version. The manuscript is in a good outstanding. Just a few minor comments:

- The introduction part is still narrow. 

- Please break the introduction into multiple paragraphs. 

- Remove the list of abbreviations from the end and place it as a table in the text. 

-Do proofreading before the final publication. 

All the best.

Author Response

(The authors gave the same response as above.)
